# Phenotyping Olive Cultivars for Drought Tolerance Using Leaf Macro-Characteristics

**Rachid Razouk \*, Lahcen Hssaini, Mohamed Alghoum, Atman Adiba and Anas Hamdani**

National Institute of Agricultural Research, Regional Agricultural Research Center of Meknes, Meknes P.O. Box 578, Morocco
* Correspondence: rachid.razouk@inra.ma

**Abstract:** This work investigates the relationships between leaf traits and olive response to water stress through two complementary field experiments in order to screen genotypes for drought tolerance. The first experiment consisted of the phenotyping of 32 olive varieties for 11 leaf morpho-physiological traits during a fruit set phase under well-watered conditions. In the second experiment, the yield and vegetative responses of five representative varieties to the withholding of irrigation during the whole fruit growth period were assessed to identify leaf traits that are associated with olive resilience plasticity and that could be used as drought tolerance markers. The results highlighted large phenotypic variability for leaf area (LA, 2.9–9.5 cm$^2$), petiole elasticity (8.0–36.0°), water loss in detached leaves (WLDL, 3.6–11.6%), stomatal density (222.6–470.1 no mm$^{-2}$), stomatal length (11.4–18.7 μm), trichome density (120.5–204.4 no mm$^{-2}$), trichome width (103.6–183.4 μm), leaf contents in cuticular wax (CWC, 44.7–606.2 μg cm$^{-2}$), and soluble sugars (SSC, 15.8–536.9 mg g$_{dw}$$^{-1}$). Among these leaf traits, WLDL displayed a significant correlation with the yield stability index (r = −0.973) and water use efficiency (r = −0.939), suggesting its use as relevant drought tolerance phenotypic marker. Likewise, LA, SSC, and CWC were singled out as second-level drought tolerance markers, being strongly correlated with stability of leaf size and leafing intensity in response to water stress. Cluster analysis highlighted 12 distinct levels of drought tolerance within the studied olive collection. Based on the four identified phenotypic markers, "Lechin de Sevilla" alongside "Picholine Marocaine" were found to be the most drought-tolerant varieties, while Frantoio was the most sensitive. This study provides the first, unprecedented, insights into the usefulness of leaf phenotyping in olive drought tolerance screening, with a focus on structural and functional leaf traits.

**Keywords:** *Olea europaea* L.; leaf traits; drought tolerance; phenotypic markers; water loss





## 1. Introduction

Drought is the most common environmental stress affecting olive production, and this stress is particularly substantial in the Mediterranean Basin where the climate is typically characterized by high potential evaporation and a low rainfall level during the growing season [1,2]. To deal with this challenge, agricultural research is focused on two main approaches: the use of improved and precise deficit irrigation management practices that are able to minimize the impact of stress on yield and vegetative growth, and the selection of plant materials that are less water-demanding or able to tolerate water stress with minimum impact on yield and fruit quality [3,4]. Both of these approaches are interesting, but the second approach seems more promising and sustainable for olive trees, as olives are generally cultivated under rainfed or deficit irrigation conditions. In addition, the selection of drought-tolerant olive cultivars is a strategic goal because of the significant adaptation of the species to arid conditions [5,6].

It is well known that the drought tolerance of plants, including olives, is determined by various genetic factors that encode morphological, phenological, physiological, biochemical, and molecular traits [7–9]. In addition, plants have developed different active mechanisms that inhibit or alleviate the harmful effects of water stress [10,11]. Olive

plants are particularly characterized by anatomical adaptations that allow them to preserve their vital functions under severe water deficit conditions. These mechanisms include leaf morpho-anatomical features that involve a dense layer of peltate trichomes that cover the abaxial leaf surface with a reduced stomatal density (200 mm$^{-2}$ to 500 stomata mm$^{-2}$), which limits water loss at the leaf level [12]. Regarding physiological mechanisms, it has been observed that olive plants can ensure a significant transpiration flow under high water potential, due to the reduced diameter of their xylemic vessels, combined with a fairly deep and branched root system [13]. Indeed, olive leaves are highly efficient and capable of ensuring photosynthesis and transpiration activity at very low levels of leaf water potential ($-6.0$ MPa) [14]. Biochemically, olive plants develop various mechanisms to survive under stressful conditions, including the accumulation of osmoprotectants, such as sugars, proteins, amino acids (proline, aspartic acid, glutamic acid) and their derived compounds, as well as enzymatic and non-enzymatic antioxidant compounds [15].

Olive trees are slow growing, with physiological and biochemical adjustments that are the predominant adaptation mechanisms to short-term water deficits, rather than morphological changes [16]. The activation of these adaptation mechanisms often affects the growth rate of leaves, shoots, and fruits, which may lead to significant yield decreases in a particular year and in following years [17]. However, the magnitude of the negative effects of water stress is not the same for all olive genotypes, due to differences in leaf structure and morphology, stomatal conductance, and water use efficiency [18,19].

In some previous works, olive cultivars' plasticity with respect to water stress have been related to the protective responses at the leaf level, especially the scavenging of reactive oxygen species by enzymatic and non-enzymatic systems, to avoid damage in photosynthesis and cell tissue [20,21]. Further investigations of the morpho-physiological and biochemical mechanisms involved in drought tolerance within olive genotypes are needed with to the goal of identifying relevant drought markers. In this regard, particular attention to early markers to simplify the screening process for the drought tolerance of large groups of olive trees would be of interest.

In this work, an ex situ collection of 32 olive varieties that are among the most cultivated in the Mediterranean Basin was screened on the basis of morphological and physiological traits that are related, directly or indirectly, to plant water status. The plants were observed during their flowering stage, which is a critical phase in production. In addition, we assessed the responses to induced water stress of five varieties of olive from different geographical origins, under field conditions, in order to identify the traits that were most correlated to the stability indices of yield components and vegetative growth. Such traits were considered as drought tolerance markers in olive and, therefore, were used in ranking the whole collection.

## 2. Material and Methods

### 2.1. Overall Approach

We performed two complementary experiments to achieve the targeted results of the present work. The first experiment consisted of screening an ex situ collection of 32 well-watered olive varieties (Table 1) on the basis of morpho-physiological and biochemical leaf traits that are commonly known to be related to water use efficiency. The targeted objectives of this experiment were to elucidate the magnitude of genotypic variation via these traits and to identify those traits that had have the highest discrimination effect. In the second experiment, we assessed the effect of induced drought stress on the yield and vegetative growth levels of five varieties of olive from different geographical origins: "Picholine Marocaine", "Arbequina", "Frantoio", "Madural" and "Sevillenca". This trial was carried out to determine the traits (identified in the first experiment) that discriminate most significantly between olive genotypes, and those traits that correlate to the stability indices of yield and vegetative growth parameters. Those traits were thereafter used as phenotypic markers in ranking the studied olive accessions with respect to drought tolerance.

**Table 1.** List of olive varieties included in the study, with their geographical origins.

| Variety | Origin | Variety | Origin |
|---------|--------|---------|--------|
| Amellau | France | Grappolo | Italy |
| Americano | Italy | Leucocarpa | Italy |
| Arbequina * | Spain | Leccino | Italy |
| Ascolana Tenra | Italy | Lechin de Sevilla | Spain |
| Azeradj | Algeria | Madonna | Italy |
| Blanqueta | Spain | Madural * | Portugal |
| Blanquette | France | Maurino | Italy |
| Cellina | Italy | Meslalla | Morocco |
| Changlot Real | Spain | Moraiolo | Italy |
| Chetoui | Tunisia | Morisca | Spain |
| Cornicabra | Spain | Ocal | Spain |
| Coratina | Italy | Piangente | Italy |
| Grappuda | Italy | Picholine Marocaine * | Morocco |
| Dritta | Italy | Sevillenca * | Spain |
| Frantoio * | Italy | Verdial | Portugal |
| Galega Vulgar | Portugal | Vernina | Italy |

*: olive varieties included in the second experiment.

The two experiments were carried out at the experimental station of the Regional Agricultural Research Center of Meknes in north-central Morocco (33°55′32″ N, 5°16′26″ W, 488 m asl). All of the olive trees were 17 years old and grew under the same edaphoclimatic conditions and agricultural practices. The climate was Mediterranean, with hot and dry summers. During the study years 2019 and 2020, the annual rainfall levels were 236 mm and 261 mm, respectively, with annual reference evapotranspiration of 1029 mm and 1094 mm, as recorded by the weather station of Ain Taoujdate at a distance of less than 100 m from the experimental plots. The soil was of loamy clay texture, moderately alkaline (pH 7.72), slightly calcareous (3.01% $CaCO_3$), and quite rich in organic matter (2.04%). Fertilization was 300 g N, 150 g $P_2O_5$, and 240 g $K_2O$ per tree, based on soil analysis, and phytosanitary treatments were managed according to local practices, mainly against fly and moth.

### 2.2. First Experiment

In the first experiment, an ex situ collection of 32 olive varieties was screened for their morpho-physiological and biochemical leaf traits during the flowering period under clear sky conditions in April. The olive varieties in the collection were planted in 2002 in parallel rows, using three replicates (trees) for each variety, with a spacing of 5 m × 3 m. The area was drip irrigated from the beginning of bud break in early March to harvest at the end of November, according to crop evapotranspiration (ETc), using two emitters per tree, delivering $4 \, L \, h^{-1}$ each. ETc was scheduled according to daily reference evapotranspiration values ($ET_0$), crop coefficients (Kc) as obtained by Fernández et al. [22], and the reduction coefficient (Kr) developed by Fereres et al. [23]. During rainy days, the effective rainfall values (80% of the recorded rainfall) were taken into account in the calculation of the irrigation water requirements. The measurements concerned 11 leaf traits: leaf area, petiole elasticity, water loss in detached leaves, stomatal density, stomatal pore length, trichome density, trichome diameter, trichomes per stomata, trichoms area index, contents in cuticular wax, and soluble sugars. All of these traits were measured on samples of fully developed leaves at midday, from the middle part of one-year-old shoots on the four positions of the tree canopy.

Leaf area (LA) and petiole elasticity (PE) were determined for samples of 60 fully-developed leaves per variety (20 leaves per replicate). LA was measured by a portable leaf area meter (AM350, ADC Bioscientific, Hoddesdon, UK). PE was measured by an assessment of the resistance of petiole to bending. To quantify this resistance, a standard weight of 0.723 g was suspended from the test leaf by a thread of 2 cm in length that was

held by a pin positioned at 4 cm from the point of the leaf attachment on the stem. Petiole elasticity was expressed in terms of the degree of the angle of the leaf curvature [24].

The water loss in the detached leaves was measured by weighing individual leaves after their exposure for 1 h to an ambient temperature of approximately 25 °C with 19% relative humidity of air. The leaves were arranged with the lower surfaces uppermost in full daylight, but the leaves were not exposed to direct sun radiation. The measurements were performed on 30 leaves per variety (10 leaves per replicate). Water loss was expressed as a percentage of the initial leaf weight immediately after its detachment [24].

Trichome traits were observed on small pieces of leaf (0.5 cm$^2$) that were removed from the center of 18 leaves per variety (six leaves per replicate). The layers of trichomes were removed from the abaxial side of the leaf pieces by transparent adhesive tape. Then, the taped pieces were mounted on a glass slide and the trichome density (TD) and width (TW) were observed, using a photomicroscope connected to a digital camera (DeltaPix, Invenio 12EIII, Horsholm, Denmark). This approach was repeated on each leaf piece until no trichome was observed on the taped piece, indicating that the whole trichome layers had been removed. The trichome area index (TAI), which corresponded to the total area covered by trichomes per mm$^2$ of leaf area, was then calculated by multiplying the trichome area ($\pi \times$ R$^2$) by TD, where R was the mean radius of the trichomes (TW/2).

Thirty leaves per variety were selected to measure the stomatal parameters based on the impression approach, as described by Gitz and Baker [25]. First, peltate trichomes were removed, using an adhesive tape, from the abaxial epidermis of the leaf. The leaf was then smeared with a thin layer of clear nail polish. Once dried, the thin polish film was removed from the leaf surface by an adhesive transparent tape and mounted on a glass slide. The impressions were observed using a photomicroscope connected to a digital camera (DeltaPix, Invenio 12EIII, Smørum, Denmark). Three areas per leaf were examined, as sub-replications, with respect to the number of stomata per mm$^2$ and the stomatal pore length.

The cuticular wax content (CWC) was determined for 10 leaves of each of six sub-samples per variety. The cuticular wax was extracted from each of the leaf sub-samples, which were previously washed with distilled water, by moving the leaves in concentrated chloroform for 30 s. The extracted wax was isolated by evaporation of the chloroform on a hot plate and weighed using precision balance 0.001 g. CWC was expressed in µg cm$^{-2}$ of the leaf area [26].

Soluble sugars were extracted according to the method of Dubois et al. [27]. Briefly, 50 mg of lyophilized leaf sample were ground in a mixture with 1 mL of 80% ethanol. The obtained extract was centrifuged at 2000 rpm for 40 min at 4 °C. Then, 0.5 mL of extract was homogenized with 0.5 mL of phenol and 1.5 mL of concentrated sulfuric acid. The mixture was heated in a water bath at 95 °C for 5 min, before measurement of the absorbance at 485 nm by a spectrophotometer (6850 UV/VIS, Jenway, Staffordshire, UK), using a glucose solution as the standard.

*2.3. Second Experiment*

In the second experiment, the response to water stress was assessed during two consecutive years (2019 and 2020) on five screened varieties: "Picholine Marocaine", "Arbequina", "Frantoio", "Madural" and "Sevillenca". These varieties, from various geographical origins, were selected on the basis of the assumption that they responded differently to water stress. They were grown as part of the collection, with 10 trees each arranged in parallel rows at the same spacing as that of the screened collection (5 m × 3 m). The technical management of this experimental plot was the same as the technical management in the collection generally, except that the irrigation was varied in order to assess the response of the varieties to drought. Water stress was induced on each variety by withholding irrigation from the fruit set at the end of May to harvest at the end of November on five trees, to produce two water treatments: well-watered trees (WT) and stressed trees (ST). Each treatment was applied on a separate subplot that contained five trees of each variety, in order to avoid interactions

of soil water between the adjacent varieties. In addition, under each treatment, only the three central trees were considered in the measurements, while the two border trees acted as buffer plants.

The measurements on well-watered trees and stressed trees involved six components of production and vegetative growth: yield (fruit and oil), average fruit weight (g), oil content ($\%_{fw}$), leaf area ($cm^2$), and leafing intensity (number of leaves per dm of one-year-old shoots). Indeed, at the ripening stage at the end of November, mature fruit samples of approximately 3 kg each were collected from 10 randomly selected fruiting branches per replicate (tree) to determinate the fruit weight. This method of fruit sampling was adopted because it takes into account fruit size variability within an individual tree. The fruit oil content was determined for previously dried and weighed fruit sub-samples, using a nuclear magnetic resonance analyzer (type Oxford 4000, Abingdon-on-Thames, UK. The fruit yield per tree was weighed in the field and the oil yield was estimated as a product of fruit yield and oil content. Leaf areas were measured on 20 full-developed sample leaves from around the canopy of a single tree (60 leaves per water treatment). The leafing intensity was determined by counting the leaves on 10 cm of 1-year-old branches (30 branches per water treatment).

### 2.4. Statistical Analysis

A Student–Newman–Keuls test (S–N–K) was performed at $p \leq 0.05$ to compare the sample means between the olive varieties and the water treatments after a prior analysis of normality test, followed by variance analysis. In order to identify the phenotypic drought tolerance markers, a correlation test was performed, using a Pearson model, between the leaf traits and the stability indices of yield, fruit weight, oil content, and vegetative parameters, which were calculated by dividing the values under water stress by the values under well-watered conditions. The identified marker traits were the basis of a two-level clustering of the varieties with respect to drought tolerance, using the S–N–K test (level 1), followed by the unweighted pair group method with an arithmetic mean (level 2). All of the statistical treatments were performed with IBM SPSS Statistics v22.

## 3. Results and Discussion

### 3.1. Traits Variability within the Collection

Large variabilities in each of the assessed leaf traits were found among the 32 studied olive varieties. The differences between the extreme values were approximately two times for the stomatal and trichomes traits, three times for leaf area (LA), four times for petiole elasticity, six times for water loss in detached leaves, 14 times for cuticular wax content (CWC), and 34 times for soluble sugars content (SSC) (Table 2). This wide range of variation showed the large diversity of structural and functional leaf traits that relate to water use efficiency. Such diversity could reflect the existence of a broad range of plasticity levels within the olive species to abiotic stress, including drought [28].

**Table 2.** Extreme values, means, and analysis of variance of the leaf traits of the 32 studied olive varieties.

| | Min | Max | Mean | Std. Deviation | Mean Square | ANOVA *p*-Value |
|---|---|---|---|---|---|---|
| Leaf area ($cm^2$) | 2.9 | 9.5 | 5.00 | 1.5 | 18.7 | <0.001 |
| Petiole elasticity (°) | 8.0 | 36.0 | 20.6 | 8.1 | 240.2 | <0.001 |
| Water loss in detached leaves (%) | 3.6 | 11.6 | 6.6 | 3.2 | 2.8 | <0.001 |
| Stomatal density (no $mm^{-2}$) | 222.6 | 470.1 | 337.9 | 58.3 | 6796.6 | 0.002 |
| Stomatal length (μm) | 11.4 | 18.7 | 16.3 | 1.7 | 5.8 | 0.001 |
| Trichomes density (no $mm^{-2}$) | 120.5 | 204.4 | 158.5 | 24.1 | 207.1 | <0.001 |
| Trichome width (μm) | 103.6 | 183.4 | 138.8 | 25.5 | 1582.3 | <0.001 |
| Trichomes per stoma | 0.3 | 0.7 | 0.5 | 0.1 | 0.03 | <0.001 |
| TAI ($mm^2\ mm^{-2}$) | 1.1 | 4.8 | 2.4 | 1.0 | 9.9 | <0.001 |
| SSC ($mg\ g_{dw}^{-1}$) | 15.8 | 536.9 | 167.3 | 138.7 | 0.05 | <0.001 |
| CWC ($\mu g\ cm^{-2}$) | 44.7 | 606.2 | 240.4 | 177.0 | 1817.3 | <0.001 |

TAI: trichomes area index; SSC: soluble sugars content; CWC: cuticular wax content.

The S–N–K test highlighted 10 distinct groups of varieties in terms of mean LA (Table 3). The lowest LA values were shown by "Sevillenca" (2.9 cm$^2$), "Blanquette" (3.1 cm$^2$), and "Picholine Marocaine" (3.4 cm$^2$). However, the highest values were recorded in "Ascolana Tenera" (7.1 cm$^2$), "Azreadj" (7.4 cm$^2$), "Madonna" (8.5 cm$^2$), and "Grappolo" (9.5 cm$^2$). Petiole elasticity (PE), with respect to which the low values indicated bending stiffness in petioles (a favorable characteristic for drought tolerance [24]) varied from 8.0° to 36.0°, with 15 dissimilar groups revealed by the S–N–K test. Based on this observation, "Meslalla", "Americano", "Grappuda", and "Vernina" seemed to be the less vulnerable varieties to leaf bending under water stress conditions, as they presented the lowest PE values (8.0° to 9.1°), while the "Blanqueta", "Lechin de Sevilla", and "Piangente" varieties were highly vulnerable, with higher PE values (34.1° to 36.0°).

**Table 3.** Mean values of leaf area (LA, cm$^2$), stomatal density (SD, n mm$^2$), stomatal length (SL, μm), trichomes density (TD, n mm$^{-2}$), trichome width (TW, μm), trichomes per stoma (TD/SD), trichome area index (TAI, mm$^2$ mm$^{-2}$), petiole elasticity (PE, °), water loss in detached leaves (WLDL, %), soluble sugars content (SSC, mg g$_{dw}$$^{-1}$), and cuticular wax content (CWC, μg cm$^{-2}$) of the 32 studied olive varieties under well-watered conditions.

| Varieties | LA | SD | SL | TD | TW | TD/SD | TAI | PE | WLDL | SSC | CWC |
|---|---|---|---|---|---|---|---|---|---|---|---|
| Amellau | 5.7 [bcde] | 338.1 [abc] | 16.1 [ab] | 174.2 [fghi] | 129.1 [cdef] | 0.5 [bc] | 2.3 [cd] | 13.1 [abc] | 5.4 [ab] | 292.3 [bcd] | 141.4 [abc] |
| Americano | 4.4 [abc] | 354.5 [abc] | 15.7 [ab] | 131.8 [abc] | 176.9 [g] | 0.4 [ab] | 3.2 [fgh] | 8.0 [a] | 4.9 [ab] | 49.3 [ab] | 105.9 [a] |
| Arbequina | 3.9 [abc] | 404.4 [abc] | 14.5 [ab] | 170.3 [fghi] | 116.1 [abcd] | 0.4 [abc] | 1.8 [abcd] | 18.0 [efg] | 6.9 [abc] | 117.7 [ab] | 163.5 [abc] |
| Ascolana Tenra | 7.1 [def] | 296.9 [abc] | 17.9 [c] | 153.2 [defg] | 119.9 [abcd] | 0.5 [bc] | 1.7 [abcd] | 28.0 [ghi] | 6.9 [abc] | 176.6 [abc] | 184.3 [bcd] |
| Azeradj | 7.4 [ef] | 370.8 [abc] | 15.0 [ab] | 124.3 [ab] | 130.3 [cdef] | 0.3 [ab] | 1.7 [abcd] | 22.1 [defg] | 4.2 [a] | 164.4 [abc] | 44.7 [a] |
| Blanqueta | 5.7 [bcde] | 429.3 [bc] | 14.4 [ab] | 194.7 [ij] | 110.5 [abc] | 0.4 [bc] | 1.9 [bcd] | 36.0 [j] | 6.2 [abc] | 62.9 [ab] | 58.4 [a] |
| Blanquette | 3.1 [a] | 354.5 [abc] | 15.3 [ab] | 162.4 [efgh] | 182.7 [g] | 0.5 [bc] | 4.3 [jk] | 17.0 [bcde] | 7.3 [bc] | 142.5 [ab] | 481.5 [cd] |
| Cellina | 4.2 [abc] | 313.3 [abc] | 17.2 [c] | 194.4 [ij] | 133.1 [cdef] | 0.6 [bc] | 2.7 [efg] | 23.1 [efg] | 5.3 [ab] | 268.0 [abcd] | 510.5 [def] |
| Changlot Real | 4.1 [abc] | 255.5 [ab] | 18.6 [c] | 167.2 [fghi] | 158.3 [defg] | 0.6 [c] | 3.3 [fgh] | 31.0 [hij] | 5.4 [ab] | 254.2 [abcd] | 451.7 [de] |
| Chetoui | 4.9 [abc] | 429.3 [bc] | 14.4 [c] | 150.4 [defg] | 103.6 [a] | 0.3 [ab] | 1.3 [ab] | 18.1 [cdef] | 5.0 [ab] | 72.5 [ab] | 483.2 [ef] |
| Cornicabra | 4.1 [abc] | 304.9 [abc] | 17.2 [c] | 138.0 [abcd] | 153.5 [defg] | 0.4 [bc] | 2.5 [def] | 13.1 [abc] | 5.4 [ab] | 42.0 [ab] | 104.2 [a] |
| Coratina | 6.3 [cde] | 304.8 [abc] | 17.3 [c] | 127.6 [abc] | 171.5 [fg] | 0.4 [abc] | 2.9 [fgh] | 18.9 [cdef] | 9.7 [de] | 230.2 [abcd] | 352.0 [ef] |
| Grappuda | 4.1 [abc] | 387.3 [abc] | 14.9 [ab] | 188.7 [hij] | 180.9 [g] | 0.5 [bc] | 4.8 [l] | 9.1 [a] | 7.1 [abc] | 487.2 [e] | 250.4 [bcd] |
| Dritta | 3.9 [abc] | 222.6 [a] | 18.7 [c] | 127.0 [abc] | 131.1 [cdef] | 0.6 [bc] | 1.7 [abcd] | 24.0 [efg] | 6.7 [abc] | 43.2 [ab] | 192.1 [bcd] |
| Frantoio | 5.2 [abcd] | 330.6 [abc] | 16.4 [ab] | 180.9 [ghij] | 126.7 [bcde] | 0.5 [bc] | 2.3 [cd] | 11.0 [ab] | 11.5 [e] | 134.4 [ab] | 339.1 [de] |
| Galega Vulgar | 4.5 [abc] | 313.6 [abc] | 17.2 [c] | 204.4 [j] | 104.0 [a] | 0.6 [c] | 1.7 [abcd] | 28.0 [ghi] | 5.3 [ab] | 536.9 [f] | 298.7 [bcd] |
| Grappolo | 9.5 [g] | 445.5 [bc] | 13.7 [ab] | 127.0 [abc] | 124.8 [bcde] | 0.3 [a] | 1.5 [abcd] | 22.0 [defg] | 7.1 [abc] | 17.7 [a] | 196.0 [de] |
| Leucocarpa | 4.7 [abc] | 371.5 [abc] | 14.9 [ab] | 176.8 [ghij] | 115.2 [abcd] | 0.5 [bc] | 1.8 [abcd] | 18.0 [cdef] | 6.7 [abc] | 57.6 [ab] | 281.5 [bcd] |
| Leccino | 5.4 [abcd] | 271.9 [abc] | 18.4 [c] | 127.0 [abc] | 107.6 [ab] | 0.5 [bc] | 1.1 [a] | 22.1 [defg] | 4.4 [ab] | 31.1 [ab] | 66.9 [a] |
| Lechin de Sevilla | 3.7 [ab] | 247.7 [ab] | 18.6 [c] | 155.7 [defg] | 164.6 [efg] | 0.6 [c] | 3.3 [ghi] | 36.0 [j] | 3.6 [a] | 386.1 [cde] | 60.7 [a] |
| Madonna | 8.5 [fg] | 330.0 [abc] | 17.1 [c] | 136.9 [abcd] | 153.3 [defg] | 0.4 [abc] | 2.5 [cde] | 11.0 [abc] | 6.7 [abc] | 280.1 [abcd] | 164.1 [cd] |
| Madural | 3.7 [ab] | 346.5 [abc] | 15.8 [ab] | 145.0 [bcde] | 142.5 [defg] | 0.4 [abc] | 2.3 [cd] | 25.1 [fgh] | 8.6 [cde] | 198.1 [abcd] | 110.5 [ab] |
| Maurino | 5.1 [abcd] | 470.1 [c] | 11.4 [a] | 178.1 [ghij] | 121.2 [abcd] | 0.4 [ab] | 2.0 [bcd] | 16.1 [bcde] | 7.0 [abc] | 167.7 [abc] | 59.3 [a] |
| Meslalla | 5.4 [abcd] | 297.0 [abc] | 17.6 [c] | 150.7 [defg] | 140.9 [defg] | 0.5 [bc] | 2.3 [cd] | 8.0 [a] | 4.6 [ab] | 90.9 [ab] | 518.8 [ef] |
| Moraiolo | 4.6 [abc] | 345.5 [abc] | 16.0 [ab] | 120.5 [a] | 149.6 [defg] | 0.3 [ab] | 2.1 [cd] | 31.1 [ij] | 10.2 [e] | 95.6 [ab] | 517.8 [ef] |
| Morisca | 5.2 [abcd] | 396.3 [abc] | 14.8 [ab] | 167.0 [fgh] | 115.8 [abcd] | 0.4 [abc] | 1.8 [abcd] | 21.1 [defg] | 5.4 [ab] | 15.8 [a] | 77.8 [a] |
| Ocal | 4.9 [abc] | 280.4 [abc] | 18.4 [c] | 197.0 [ij] | 108.9 [abc] | 0.7 [c] | 1.8 [abcd] | 15.0 [abcd] | 7.3 [bc] | 47.5 [ab] | 488.0 [ef] |
| Piangente | 3.9 [abc] | 337.8 [abc] | 16.1 [ab] | 157.9 [defg] | 166.9 [efg] | 0.5 [bc] | 3.4 [hij] | 34.1 [j] | 5.1 [ab] | 88.1 [ab] | 112.9 [ab] |
| Picholine Mar. | 3.4 [a] | 288.6 [abc] | 18.3 [c] | 170.7 [fghi] | 166.1 [efg] | 0.6 [bc] | 3.7 [ijk] | 25.0 [fgh] | 3.8 [a] | 414.1 [de] | 107.3 [a] |
| Sevillenca | 2.9 [a] | 305.0 [abc] | 17.2 [c] | 150.1 [defg] | 142.2 [defg] | 0.5 [bc] | 2.4 [cde] | 23.0 [efg] | 9.6 [de] | 93.0 [ab] | 79.8 [a] |
| Verdial | 4.0 [abc] | 321.5 [abc] | 17.1 [c] | 175.6 [fghi] | 183.4 [g] | 0.5 [bc] | 4.6 [kl] | 23.0 [efg] | 5.2 [ab] | 235.5 [abcd] | 83.8 [a] |
| Vernina | 5.1 [abcd] | 346.5 [abc] | 16.0 [ab] | 146.8 [cdef] | 109.5 [abc] | 0.4 [abc] | 1.4 [abc] | 9.1 [a] | 11.5 [e] | 62.5 [ab] | 606.2 [g] |

The values marked within columns by different letters are significantly different at *p* < 0.05.

A higher stomatal density is considered to be an undesirable criterion for drought tolerance, as it increases transpiration loss, which is considered to be a limiting factor under water deficit conditions [29]. According to the S–N–K test (Table 3), the "Maurino" variety was distinguished by the highest number of stomata per mm$^2$ (470.1), followed by "Grappolo" (445.5 stomata mm$^{-2}$), "Blanqueta", and "Chetoui" (429.3). The lowest stomata density was observed in "Dritta" (222.6), "Lechin de Sevilla" (247.7), and "Changlot real" (255.5). In the other varieties, the stomatal density values were statistically similar, varying within a very tiny interval with an overall average of 352.3 stomata mm$^{-2}$. The stomata density was negatively correlated to stomata size, as previously found in several

plants, including olive [30,31]. The control of stomata closure is a complex trait that is affected by many internal and external factors. The control is known to be faster for smaller stomata, indicating that varieties with small stomata could be more tolerant to drought [32]. The smallest stomata were found in "Maurino", with an average length of 11.4 μm, while the largest stomata were observed in "Dritta" (18.7 μm). Trichomes are more abundant and larger, and therefore better for drought tolerance, due to their role in regulating transpiration at the leaf surface [33,34]. Unlike the stomatal traits, no significant correlation was found between the density of trichomes and their size. Within the studied varieties, the number of trichomes per $mm^2$ varied from 120.5 in "Moraiolo" to 204.4 in "Galega vulgar", with an average of 158.51, while trichome diameters ranged between 103.61 μm in "Chetoui" and 183.4 μm in "Verdial", with an average of 138.8 μm. Trichomes are arranged from a single layer up to five layers, depending on varieties, as shown by the TAI values, at a rate of 0.3 to 0.7 trichomes per stoma.

Cuticular wax and soluble sugars are among the biochemical compounds whose accumulation in leaves is associated with water stress resistance in plants [16]. The cuticular wax act as a photoprotective layer and soluble sugars are the osmoregulators. Both contribute in maintaining leaf water potential under drought stress [35]. In the studied olive collection, variations in CWC and SSC were highly significant ($p < 0.001$), with extreme values of 44.73 μg $cm^{-2}$ to 606.20 μg $cm^{-2}$ and 15.8 mg $g_{dw}^{-1}$ to 536.9 mg $g_{dw}^{-1}$, respectively. However, it is important to emphasize that these initial values of the biochemical traits may be associated with drought tolerance in the short term, whereas in the medium and long terms their accumulation rates are rather more determinant in assessing the variety's adaptation to water stress [36,37].

Moreover, previous studies have suggested that leaf conductance, assessed in the present work by water loss in detached leaves, is one of the potential screening criteria for the drought tolerance of plants [38,39]. Leaf conductance indicates a plant's ability to control water loss in leaves under drought conditions, both through the stomata and the cuticle [40]. The S–N–K test revealed seven distinct groups of varieties within the studied collection regarding this trait ($p < 0.001$). The lowest values, indicating the highest leaf resistance to water loss, were shown by "Lechin de Sevilla", "Azeradj", and "Picholine Marocaine", with an average value of 3.9% during 1 h. However, the highest values were recorded in the leaves of "Frantoio", "Vernina", and "Moraiolo", with an average value of 11.1%, showing the weakest ability to control water loss.

### 3.2. Olive Varieties Response to Drought

There were significant differences among the five tested olive varieties with respect to their adaptation to the environment of the experimental site. This was particularly revealed by the genotypic variations in yield and leafing intensity for the irrigated trees, which showed that "Picholine Marocaine" was the most efficient variety under full irrigation, followed by "Sevillenca" and "Arbequina", with yield levels in the range of 20.7 kg $tree^{-1}$ to 26.8 kg $tree^{-1}$ for the two consecutive experimental years (2019 and 2020). In contrast, "Madural" and "Frantoio" seemed less efficient, with yield levels less than 50% when compared to "Picholine Marocaine" (Table 4). This variability in the behavior of olive varieties is related to differences in non-hydric factors that affect their rates of flowering and the fruit set, such as chilling and heating requirements [41]. Under rainfed conditions, "Picholine Marocaine" remained significantly efficient with a yield average of 17.7 kg $tree^{-1}$. "Madural" and "Frantoio" were less efficient, yielding an average of 2.4 kg $tree^{-1}$ and 1.8 kg $tree^{-1}$, respectively, for the two experimental years. However, "Arbequina" was more productive than "Sevillenca"; both were assessed as moderately efficient, compared with the other varieties.

**Table 4.** Yield (kg tree$^{-1}$), fruit weight (g), oil content (%$_{fw}$), leaf area (cm$^2$), leafing intensity (leaves dm$^{-1}$), and water use efficiency for fruit (WUE$_f$, kg m$^{-3}$) and oil (WUE$_o$, kg m$^{-3}$) production per tree in watered trees (WT) and stressed (ST) trees of the studied olive varieties for the two experimental years.

| Year | Variety | | Fruit Yield | Oil Yield | Fruit Weight | Oil Content | Leaf Area | Leaves dm$^{-1}$ | WUE$_f$ | WUE$_o$ |
|------|---------|----|-------------|-----------|--------------|-------------|-----------|---------|---------|---------|
| 2019 | Picholine | WT | 28.6 d | 4.0 c | 5.1 c | 13.9 ab | 5.7 de | 14.3 c | 3.1 bc | 0.4 b |
| | | ST | 19.5 c | 3.4 c | 4.7 c | 17.6 bc | 5.3 c | 5.7 a | 5.3 d | 0.9 d |
| | Arbequina | WT | 21.4 c | 3.2 c | 1.7 a | 14.8 ab | 6.6 g | 11.0 b | 2.3 b | 0.3 b |
| | | ST | 13.3 b | 2.5 bc | 1.7 a | 19.1 c | 6.0 ef | 8.0 ab | 3.6 c | 0.7 c |
| | Sevillenca | WT | 24.1 cd | 3.5 c | 2.2 a | 14.4 ab | 4.8 b | 10.0 ab | 2.6 bc | 0.4 b |
| | | ST | 9.4 b | 1.4 ab | 1.7 a | 15.4 abc | 4.3 a | 9.7 ab | 2.6 bc | 0.4 b |
| | Madural | WT | 3.9 a | 0.4 a | 3.9 b | 11.8 a | 6.3 fg | 7.3 ab | 0.4 a | 0.1 a |
| | | ST | 1.2 a | 0.2 a | 2.2 a | 14.5 ab | 5.4 cd | 5.0 a | 0.3 a | 0.1 a |
| | Frantoio | WT | 11.9 b | 1.8 b | 2.1 a | 15.2 abc | 8.9 h | 9.3 ab | 1.3 a | 0.2 ab |
| | | ST | 0.6 a | 0.1 a | 1.3 a | 19.4 c | 5.1 bc | 6.7 ab | 0.2 a | 0.03 a |
| 2020 | Picholine | WT | 25.0 e | 3.6 d | 5.1 c | 14.5 abc | 5.6 d | 14.7 f | 2.8 cd | 0.4 c |
| | | ST | 16.0 cd | 2.3 bcd | 4.7 c | 14.6 abc | 5.1 bc | 5.3 ab | 4.3 e | 0.6 d |
| | Arbequina | WT | 20.0 de | 3.1 cd | 1.7 a | 15.1 abc | 6.5 e | 11.3 e | 2.3 bcd | 0.3 bc |
| | | ST | 11.7 bc | 2.2 abcd | 1.7 a | 18.8 c | 5.8 d | 7.3 bcd | 3.2 d | 0.6 d |
| | Sevillenca | WT | 21.8 de | 3.4 cd | 2.2 a | 14.78 abc | 4.8 b | 10.3 e | 2.5 bcd | 0.4 bc |
| | | ST | 6.7 ab | 1.2 ab | 1.7 a | 18.6 bc | 4.2 a | 8.7 cde | 1.8 abc | 0.3 bc |
| | Madural | WT | 6.4 ab | 0.7 ab | 4.0 b | 11.7 a | 6.2 e | 7.7 bcd | 0.7 a | 0.1 a |
| | | ST | 3.6 a | 0.5 a | 2.3 a | 13.8 ab | 5.2 c | 4.3 a | 1.0 a | 0.1 ab |
| | Frantoio | WT | 13.3 c | 1.8 abc | 2.1 a | 13.9 ab | 8.8 f | 9.7 de | 1.5 ab | 0.2 abc |
| | | ST | 3.0 a | 0.5 a | 1.3 a | 16.9 bc | 4.9 bc | 6.3 abc | 0.8 a | 0.1 ab |
| | Water treatment (T) | | *** | *** | *** | *** | *** | *** | *** | *** |
| | Genotype (G) | | *** | *** | *** | * | * | ns | *** | *** |
| | Year (Y) | | * | * | ns | * | ns | ns | * | * |
| | T × G | | *** | * | *** | ns | *** | *** | *** | *** |
| | T × Y | | ns | Ns | ns | ns | ns | ns | ns | ns |
| | G × Y | | ns | Ns | ns | ns | ns | ns | ns | ns |
| | T × G × Y | | ns | Ns | ns | ns | ns | ns | ns | ns |

Mean values marked by different letters, within column for each year, are significantly different at $p \leq 0.05$; *,***: significant ANOVA at $p < 0.01$ or $p < 0.001$, respectively; ns: not significant ANOVA

Fruit yield fluctuated between the two experimental years. Yield alternation is a normal phenomenon in olive, which is characterized by a succession of a year with high yield (an on-year) and single or several years with a year of low yield (an off-year). This phenomenon results from complex interactions between a set of factors related to balance between vegetative growth and the fruiting, environment (e.g., climate and soil) and agricultural practices [42]. In our study, the first experimental year (2019) was an on-year for "Picholine Marocaine", "Arbequina", and "Sevillenca", and an off-year for "Madural" and "Frantoio"; the second year (2020) was an off-year for "Picholine Marocaine", "Arbequina", and "Sevillenca", and an on-year for "Madural" and "Frantoio". The differences in fruit yield between the two years were partly associated with water stress, with an overall average of 41% under rainfed conditions compared with 15% under full irrigation conditions (Table 4). These values fell within the range of 15% to 35% for irrigated orchards and 13% to 50% for rainfed orchards, as reported by Hadiddou et al. [43] for northern Morocco. Among the studied varieties, the "Madural" yield alternated less under water deficit condition, with a difference of approximately 7% between the two years of the study, while the "Frantoio" yield alternated with a difference of more than 80%.

Based on the WUE values under rainfed conditions, we deduced that "Picholine Marocaine" was the most drought-tolerant variety, followed in descending order by "Arbequina", "Sevillenca", "Madural", and "Frantoio". However, it is important to emphasize that these drought tolerance levels of the varieties were influenced by their degrees of adaptation under local climatic conditions. Therefore, the ranking of these varieties for drought tolerance based on WUE values may change if the same experiment were conducted under other environmental conditions. Hence, to establish a precise ranking of

the varieties, it is necessary to rely on the magnitude of the changes that are induced by water stress, using the stability indices of traits that correspond to their percentages of decrease/increase during normal watering.

The stressed trees showed a decrease in fruit yield that ranged, on average, between 33.9% in "Picholine Marocaine" and 86.3% in "Frantoio" for the two experimental years, compared with well-watered trees (Figure 1). In "Arbequina", the yield decrease mainly resulted from fruit drop, as its fruit weight remained statistically unaffected. However, in the other varieties, the fruit weight decreased and significantly contributed to yield decrease, in combination with fruit drop. Therefore, fruit drop was a common reaction of all of the tested varieties to applied water stress, diminishing or even maintaining the water stress effect on fruit weight. In this sense, Mezghani et al. [44], in Tunisia, reported that fruit weight was maintained in three olive varieties (Coratina, Manzanilla, and Chemlali), while increasing in two other varieties (Picholine and Chetoui) in the presence of a significant tree load reduction in response to water stress. Therefore, these findings suggested that the known opposite correlation between tree load and fruit weight could be non-significant under water deficit conditions, depending on the genotype used. Nevertheless, our results agree with those of Tognetti et al. [45] on "Frantoio" and those of Iniesta et al. [46] on "Arbequina". A similar trend was observed for oil yield in response to water stress, although fruit oil content increased differently in all of the studied varieties. However, the decrease rates for oil yield were somewhat less than those of fruit yield, with a difference ranging from 3.1% (Frantoio) to 15.3% (Arbequina).

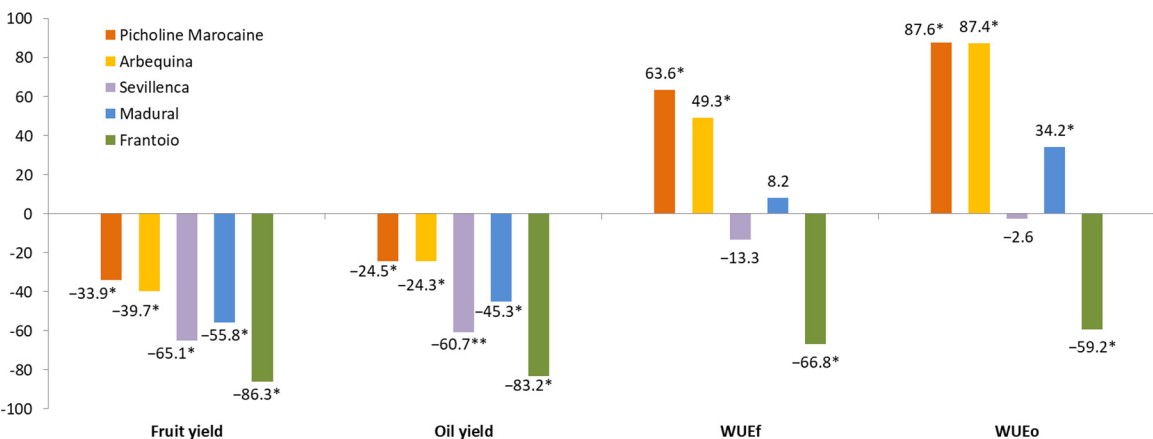

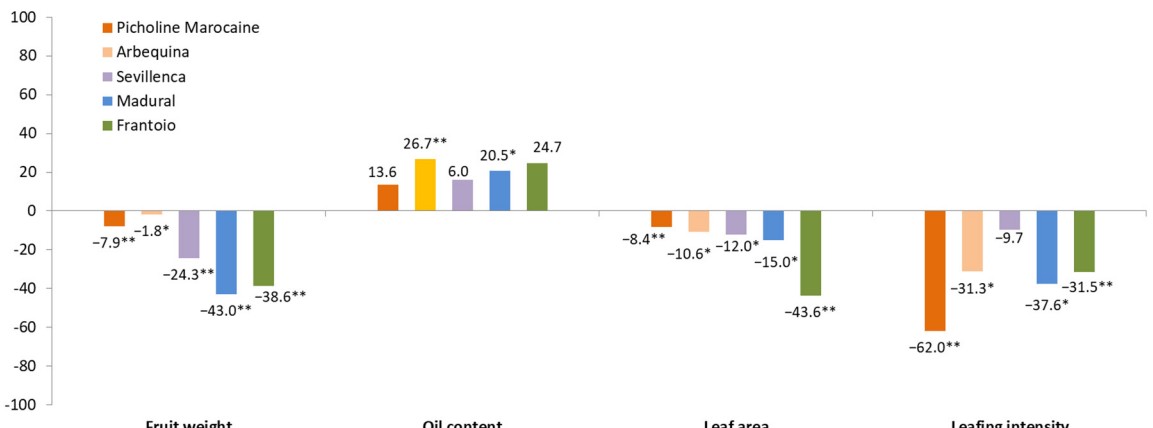

**Figure 1.** Response of Picholine Marocaine, Arbequina, Sevillenca, Madural, and Frantoio to applied water stress (averages for two experimental years 2019 and 2020). *,**: significant decrease/increase at $p < 0.05$ or $p < 0.01$, respectively; WUE: water use efficiency for fruit (WUE$_f$) and oil production (WUE$_o$).

Moreover, the impact of water stress on olive yield seemed to be higher during an off-year. Indeed, the yield decrease during the off-year was higher, from 3.4% (Arbequina) to 26.4% (Madural), during the on-year (Table 4). Such olive responses to water stress were previously observed in similar works on "Frantoio", "Leccino" [45], "Cobrançosa" [47], and "Coratina" [14]. This indicates that water stress amplifies the imbalances between vegetative growth and fruiting, which are already induced by the high fruit load that is recorded during the on-year [48]. WUE in the rainfed trees was higher for "Picholine Marocaine", "Arbequina", and "Madural" by averages of 1.9 kg m$^{-3}$, 1.1 kg m$^{-3}$, and 0.1 kg m$^{-3}$, respectively, for fruit yield and 0.4 kg m$^{-3}$, 0.3 kg m$^{-3}$, and 0.03 kg m$^{-3}$, respectively, for oil production, compared with irrigated trees. In contrast, WUE was lower in "Frantoio" and "Sevillenca" by averages of 0.3 kg m$^{-3}$ and 0.3 kg m$^{-3}$ for fruit and 0.01 kg m$^{-3}$ and 0.1 kg m$^{-3}$ for oil yield, respectively. Theses variations of WUE in response to water stress suggest that "Sevillenca" is more drought sensitive than "Madural", contrary to their ranking obtained by comparing their WUE values under rainfed conditions. In fact, Sevillenca was more productive under the local environment, but showed a lower stability in its response to water stress compared with that of "Madural". This suggests that the latter variety may be more productive than "Sevillenca" under rainfed conditions in other environments, to which it is more adapted.

Fruit weight was significantly decreased by water stress in the studied varieties, except in "Arbequina" (Figure 1). "Picholine Marocaine" showed a low fruit reduction in response to water stress, with an average of 7.9% for the two years of the study, thereby confirming its ranking as the most drought-tolerant variety. In the other varieties, fruit weight decreased highly, at 24.3% for "Sevillenca", 38.6% for "Frantoio", and 43.0% for "Madural". Decreased fruit weight is an obvious response to severe water stress, although fruit drop may offset this effect and render it non-significant (Gucci et al., 2007), as observed herein with respect tyo "Arbequina". Fruit oil content showed a significant increase under rainfed conditions. Water stress can increase oil content either directly through activation of some of the genes involved in oil biosynthesis, or indirectly by reducing vegetative growth, thereby improving the light reception of trees and accelerating fruit ripening [49,50]. Nevertheless, the increase rates for oil content were similar for all varieties, with an average of approximately 20%. Hence, the genotypic variations in this trait do not seem to be indicative for the ranking of the studied varieties with regard to drought tolerance. On the other hand, water stress significantly decreased leaf area and accentuated leaf fall in all varieties, which is in accordance with previous studies [51]. The lowest decrease in leaf area (LA), of approximately 10%, was observed in "Picholine Marocaine" while in contrast this variety exhibited the highest leaf fall in response to drought stress. Therefore, the high productivity of this variety was related to its leaf load, which, despite being reduced by water stress, remained sufficient to stabilize yield level. The highest LA decrease, of approximately 44%, was recorded in "Frantoio", which may partly explain why this variety was the least productive under rainfed conditions.

Overall, the studied varieties showed different levels of sensitivity to drought. The changes induced were very significant for all of the measured parameters since the first year of the experiment. This was most probably related to the fact that the roots were more developed in the topsoil layer, as the trees were usually drip irrigated, which made them more sensitive to water stress [52]. The responses of the five varieties were fairly widespread, suggesting the possibility of identifying phenotypic markers of drought tolerance.

### 3.3. Correlations

Due to the wide variation observed in leaf traits, the 32 olive varieties studied in the first experiment displayed distinct levels in transpiration regulation. This indicated a large diversity within the collection in controlling water stress. However, each trait predicted a specific ranking of the varieties with respect to water stress tolerance. For this reason, it was essential to first determine the traits that are strongly related to the change induced by water stress. Hence, a correlation test was performed for the five olive varieties studied in

the second experiment. This test concerned leaf traits and stability indices of yield, WUE, fruit weight, leaf area, and leafing intensity in response to the applied water stress (Table 5). The correlation matrix showed that high leaf resistance, corresponding to low water loss from leaves, seemed to weaken the effect of water stress on fruit and oil yield levels, while increasing WUE. The use of this trait as a relevant indicator in assessing plant response to drought was reported in previous studies on various plants, including olive [30], maize [53], grapevines [54], and Wedelia trilobata [40].

**Table 5.** Correlation coefficients between leaf traits and stability indices (TSIs) of fruit yield, oil yield, fruit weight, oil content, leaf area, leafing intensity, and water use efficiency for fruit ($WUE_f$) and oil production ($WUE_o$) of Picholine Marocaine, Arbequina, Sevillenca, Madural, and Frantoio in response to water stress.

| Leaf Traits | TSI | | | | | | | |
| --- | --- | --- | --- | --- | --- | --- | --- | --- |
| | Fruit Yield | Oil Yield | Fruit Weight | Oil Content | Leaf Area | Leafing Intensity | $WUE_f$ | $WUE_o$ |
| Leaf area | −0.647 | −0.570 | −0.375 | 0.669 | **−0.881 *** | −0.132 | −0.647 | −0.570 |
| Stomatal density | 0.068 | 0.127 | 0.203 | 0.462 | 0.025 | 0.213 | 0.068 | 0.127 |
| Stomatal length | 0.113 | 0.059 | −0.010 | −0.376 | 0.087 | −0.340 | 0.113 | 0.059 |
| Trichomes density | −0.049 | 0.001 | 0.325 | 0.651 | −0.512 | −0.404 | −0.049 | 0.001 |
| Trichome width | 0.340 | 0.311 | 0.030 | −0.247 | 0.388 | −0.538 | 0.340 | 0.311 |
| Petiole elasticity | 0.593 | 0.558 | 0.167 | −0.400 | 0.845 | −0.202 | 0.593 | 0.558 |
| Trichomes per stoma | −0.030 | −0.044 | 0.080 | 0.054 | −0.297 | −0.486 | −0.030 | −0.044 |
| Trichomes area index | 0.383 | 0.373 | 0.201 | 0.004 | 0.254 | −0.719 | 0.383 | 0.373 |
| WDLD | **−0.973 ** ** | **−0.939 *** | −0.857 | 0.153 | **−0.918 *** | 0.238 | **−0.973 ** ** | **−0.939 *** |
| SSC | 0.503 | 0.534 | 0.282 | 0.324 | 0.329 | **−0.916 *** | 0.503 | 0.534 |
| CWC | −0.674 | −0.619 | −0.328 | 0.528 | **−0.930 *** | −0.021 | −0.674 | −0.619 |

TSI: trait stability index = the value in stressed trees divided by the value in well-watered trees; *,**: significant correlation coefficient at $p < 0.05$ or $p < 0.01$, respectively; significant correlation coefficients are marked in bold.

Water loss in detached leaves is linked to stomatal closure time and cuticular transpiration. Stomatal closure is the most common mechanism in water loss control at the leaf level, and is known to be faster for genotypes with small stomata, due to the ease of controlling the turgidity of guard cells [55]. However, although the cuticular transpiration is tiny under normal leaf hydration, it often rises under drought conditions to significant levels, depending on the histological features of the leaves of the genotypes [56]. Some works on olive highlighted that the thickness of palisade parenchyma is the histological component that is most related to water loss in leaves, while differentiating between drought-tolerant and sensitive olive cultivars [18,57]. Based on these relationships, we deduced that water loss in detached leaves encompasses changes in both leaf anatomy and gas exchange, thereby suggesting its use as a simple, rapid, and accurate method in assessing water stress tolerance at leaf scale.

In addition, a strong and negative correlation was found between leaf area and a leaf's stability index in response to water stress. Therefore, leaf area was less reduced by water stress in varieties that are characterized by small leaves, suggesting that leaf area is an indicator of drought tolerance in olive. In other studies, leaf area was generally not correlated with osmotic potential, but was known to contribute to drought tolerance by reducing water loss [58,59]. On the other hand, the correlation matrix showed that the water stress effect on leaf area was lower in olive varieties in which the leaves showed a low level of soluble sugars (SSCs) under irrigated conditions. The causal links between these two traits remain unclear; to the best of our knowledge, the links have not been previously addressed. However, some works reported that the level of SSCs in leaves is often lower in stressed olive trees compared with well-watered ones, which could partly explain this significant correlation with leaf area [60,61]. Another significant correlation was found between cuticular wax content (CWC) and the stability index of leafing intensity, indicating that leaf fall in response to water stress was less pronounced in varieties that showed a low initial level of CWC before applying water stress. One of the most probable explanations of

this correlation is that gas exchange and photosynthesis in this last category of varieties remain at efficient levels during the initial phases of water stress, due to the weakness of the wax layer in their leaves [33]. This result suggests that the initial level of CWC is a relevant indicator in assessing olive plasticity to drought, although similar studies have reported that it was, rather, the increased rate of CWC in response to water stress that was more decisive [16].

### 3.4. Hierarchical Clustering

Given the aforementioned correlations, water loss in detached leaves (WLDL) is distinguished as a first-order phenotypic drought tolerance marker, as it was related to water stress effects on the yield and WUE levels that matter most in assessing olive resilience to drought. However, leaf area, SSC, and CWC appear to be second-order phenotypic markers, as they showed significant correlations with water stress effects on vegetative growth, which could indicate a degree of olive drought tolerance in the long term. Therefore, to highlight dissimilarities among the 32 olive varieties with regard to drought tolerance, they were first distributed over seven main clusters resulting from the S–N–K test on the WDLD values (the clusters are marked by different letters in Table 3). Then, the unweighted pair group method with arithmetic mean (UPGMA) was applied on the leaf area, SSC, and CWC values within each main cluster (Figure 2).

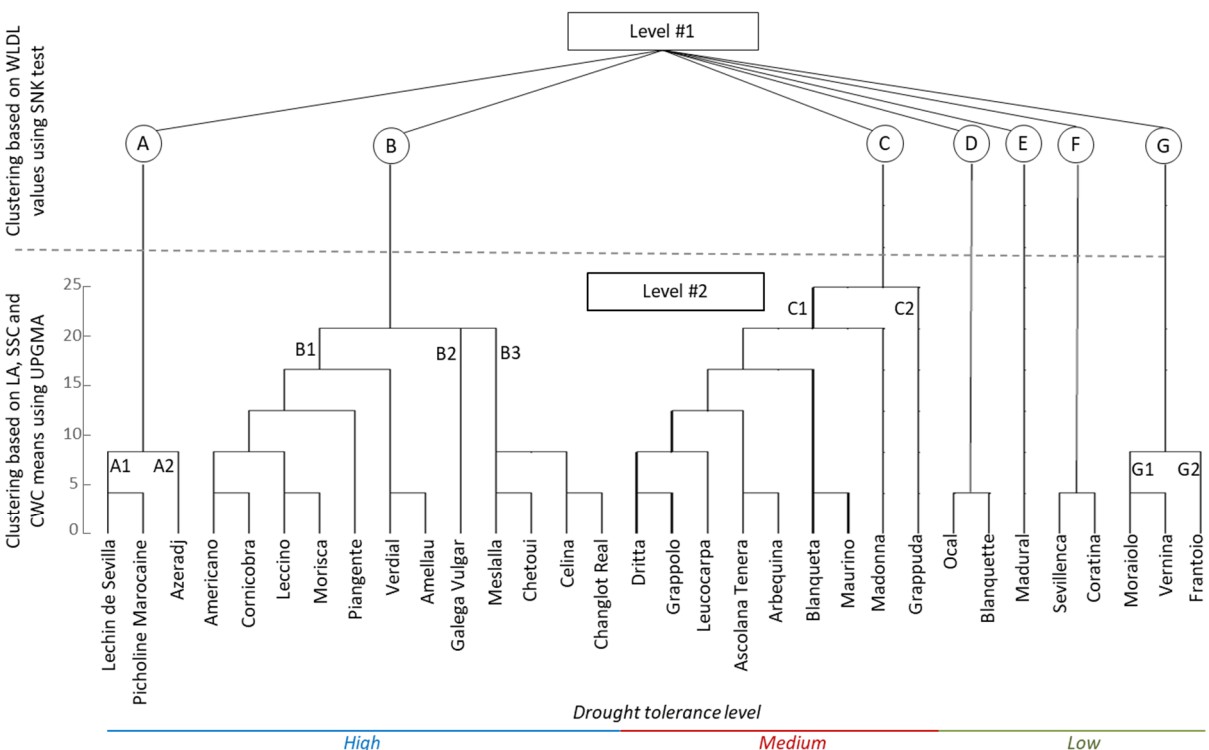

**Figure 2.** Two-level clustering of the 32 olive varieties studied in the first experiment for drought tolerance based on phenotypic markers identified by correlation test: WLDL (level 1), LA, SSC and CWC (level 2).

The first main cluster in Figure 2 (A) is formed by "Picholine Marocaine", together with "Lechin de Sevilla" and "Azeradj", which display the lowest WDLD values at an average of 3.9%. Therefore, these varieties were the most drought-tolerant ones within the studied collection, with a certain superiority for "Picholine Marocaine" and "Lechin de Sevilla" ranked in a separate sub-cluster (A1), due to their smaller leaves (3.6 cm$^2$), compared with the leaves of "Azeradj" (7.4 cm$^2$). These two varieties also showed a low density of stomata (268 stomata mm$^{-2}$) covered by large trichomes (165 mm), which constituted an additional

advantage for their adaptation to drought conditions. The second main cluster in Figure 2 (B) contains 12 varieties, including "Leccino", "Meslalla", "Chetoui", and "Cornicobra". The drought-tolerance level of this group was assessed as high, somewhat close to that of the first main cluster, as all varieties listed therein showed a higher leaf resistance than that of "Arbequina", for which the response to water stress was assessed as intermediate. Seven varieties within cluster (B), "Leccino", "Morisca", "Verdial", "Cornicobra", "Americano", "Piangente", and "Amellau", were distinguished as the most drought-tolerant. They were grouped into a distinct sub-cluster (B1), mainly due to their low CWC. The third main cluster (C) comprises 9 varieties including "Arbequina", thus displaying a medium level of drought tolerability. Among these varieties, seven were clustered alongside "Arbequina", while "Grappuda" ranked into a distinct sub-cluster (C2) as being relatively more drought sensitive due to its high leaf SSC. The fourth main cluster (D) is formed by the two varieties, "Ocal" and "Blanquette", which showed higher drought sensitivity level compared to "Grappuda", but lower than that of "Madural" ranked alone into the fifth main cluster (E). "Coratina" variety was clustered alongside "Sevillenca" in a distinct main cluster (F), displaying high sensitivity to drought compared with "Madural". However, "Moraiolo", "Vernina" and "Frantoio" are clustered into the less drought sensitive main cluster (G) displaying the highest WLDL values of 11.1% in average. Among these three varieties, "Frantoio" showed the largest leaves and highest SSC level, thereby classifying it in a separate sub-cluster (G2) as the most sensitive to drought within the studied olive collection.

For most varieties, the obtained ranking for drought tolerance was in line with the classification provided in the international olive databases that are available online at www.oleadb.it. In addition, the ranking confirmed the results reported in some previous works regarding some of the varieties studied herein, such as "Leccino" and "Frantoio" in the studies by Tognetti et al. [45] and Hadiddou et al. [43], "Arbequina" and "Blanqueta" in the study by Bacelar et al. [62], and "Lechin de Sevilla", "Arbequina", "Changlot Real", and "Blanqueta" in the study by Marin et al. [63]. The phenotypic clustering of the 32 olive varieties is very useful in deepening the analysis of the functional and structural traits related to olive drought adaptation. It is of great interest in guiding varietal choice and olive diversification in arid lands. For example, the "Lechin de Sevilla" and "Azeradj" varieties can be studied, together with "Picholine Marocaine" in Morocco or "Leccino" in Italy.

## 4. Conclusions

Water loss in detached olive leaves, measured on trees grown under well water conditions, was identified as a phenotypic marker for the drought tolerance of olive, due to its significant relationship with yield and water use efficiency responses to drought. Leaf area, as well as soluble sugars and cuticular wax contents, were found to be potential secondary drought markers that are related only to the stability of vegetative growth in response to water stress. Using these marker traits, 32 olive varieties in an ex situ collection were clustered on the basis of drought tolerance. Among them, "Lechin de Sevilla" and "Azeradj" were found to be the most drought-tolerant varieties, together with "Picholine Marocaine", which seemed promising for olive diversification in dry areas. However, "Moraiolo", "Vernina", and "Frantoio" were the most drought sensitive varieties. The drought markers identified in the present study may be of great use in further studies of larger olive collections, with a focus on selecting representative varieties for in-depth investigations of the molecular determinisms that are involved in olive drought tolerance.

**Author Contributions:** R.R.: supervision, conceptualization, methodology, investigation, data curation, software, resources, validation, review and editing, in vitro assays methodology and experiments, and writing—original draft. L.H.: formal analysis. M.A.: visualization. A.A. and A.H.: data curation. All authors have read and agreed to the published version of the manuscript.

**Funding:** This research received no external funding.

**Informed Consent Statement:** Not applicable.

**Data Availability Statement:** Not availability.

**Acknowledgments:** The authors thank M. Lahlou, A. Sirouni, and H. El Haik for their assistance in the field and laboratory work.

**Conflicts of Interest:** The authors declare no conflict of interest.

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
