# Peer review of "Phenotyping Olive Cultivars for Drought Tolerance Using Leaf Macro-Characteristics"

_horticulturae, doi:10.3390/horticulturae8100939_

Round 1
Reviewer 1 Report
Comments and Suggestions for Authors
The authors discuss Phenotyping olive cultivars for drought tolerance using leaf macro-characteristics. This topic is highly significant since drought significantly impacted crops productivity worldwide and identifying/characterizing major strategies for drought resistance will greatly assist in overcoming the great losses encountered every year.
There are critical points that should be taken into consideration by authors:
1. Abstract is descriptive where results are generally described (higher, lower... etc) no values/numbers whatsoever to describe results were found.
2. The objective of the study is not well-identified in the introduction section. What is the basis of the selection of the 32 olive varieties in 1st experiment and the 5 vars in the second experiment?
3. The basis for selection of the 32 olive verities should be identified. Their sources (origion) and major var. characteristics should be also provided. Furthermore, vernalization requirements (if available) and any other specific well known var characteristic that might alter their performance under drought should be reported in methodology?
4. It is not clear how that 32 different varieties of olives trees of 17 years old has been arranged in a completely randomized design of 3 replicates each? (The olive collection was planted following a randomized complete block design with three replicates ….). A diagram or photo describe arrangement of trees in 1st experiment is required.
5. Methodology used to measure various leaf indices should report names of devices/tools used for these measurements with their company names and country?
6. Selection of leaves for sampling in 1st experiment should be better described in methodology. It is not enough to mention for example “samples of 60 fully developed leaves per variety (20 leaves per replicate)” for leaf area and PE measurements or “measurements were performed on 30 leaves per variety (10 leaves per replicate)” for measurement of water loss rate, or “center of 18 leaves per variety” for trichomes or “30 leaves per variety” for stomata density. Sampling of leaves in a 17-year-old trees should be better explained: for example: the age of twigs/branches that has been taken from (current year growth or 1 year old?) should be specified, the position of branches/twigs used for leaf sampling, time of the day for leaf sampling, especially for those used for estimation of water loss in detached leaves. Leaf age and position within the canopy as well as time of the day at sampling might contribute to the variation obtained within the results and should be justified whether it’s a genotypic characteristic or just a response to variation in microclimate within canopy.
7. The basis and reasons for selection of varieties used in the second experiment is not reported. Should specify the reason for that: whether it is based on their performance in the 1st experiment or earlier based on reported data or recommendations?
8. Induction of water stress in the second experiment should be further explained. Length of water withholding (from May to ......)? the amounts of irrigation water applied is based on what? The quantity of water applied (cubic meter per tree of each treatment for the whole season or during the experimental period?
9. Results sections contains Invalid conclusions such as those found in section 3.1 “These wide variations may reflect the existence of a broad range of plasticity levels within the olive species to abiotic stress, including drought” since these results are obtained from non-stressed trees (1st experiment)? Another point is “A higher stomatal density is considered as a desirable criterion for drought tolerance since it ensures leaf cooling” neglecting the problem associated with higher stomatal density through increasing transpiration losses which is considered limiting factor under water deficit conditions? Another point is “The control of stomata closure is known to be faster for smaller ones, indicating that the varieties with small stomata could be more tolerant to drought [32]” neglecting that stomatal closure is a complex trait affected by many internal and external factors. Genotypic variation and hormonal regulations play more role in their responses to environmental stimuli.
10. Results of second experiment (Section 3.2 Olive varieties response to drought) indicated 2 years data which was not explained in methodology. Furthermore, variability in olive varieties behavior in section 3.2 were related to “differences in chilling and heat requirements, and therefore in the rates of flowering and fruit set”
11. Hierarchical clustering dose not clearly explain the reason for having 7 distinct groups based on SNK test.
12. Table formatting needs revising with numbers that should be reported with the minimal number of decimals that indicate proper significant figures. Tables and figures captions should also be fully explanatory describing all abbreviations used within it.
13. It is important to identify in conclusion that phenotyping of leaves for drought tolerance indices were done in trees grown under well water conditions in 1st expt. and not under water stress conditions which might affect results obtained and therefore withdrawn conclusions since environmental conditions such as water deficit stress greatly alter leaf phenotypic characteristics. The order of significant phenotypic marker for olive drought tolerance might be altered or reversed? Especially those of leaf cuticular waxes and stomatal size and density and trichomes thickness and density.
14. Authors must correct some typos in the text

Author Response
We were glad to receive your valuable comments and suggestions. Accordingly, the manuscript has undergone rigourous review and amendements. All changes were highlighted in yellow.
Please find in the attached file a point-by-point response to your comments and concerns. Thank you!

Reviewer 2 Report
The paper “Phenotyping olive cultivars for drought tolerance using leaf macro-characteristics” deals with assessing the drought tolerance of 32 olive cultivars from the ex-situ collection of the Regional Agricultural Research Centre of Meknes, Morocco. The experiment is well planned and executed. The paper is well written but there are several major issues which need to be addressed prior to publication:
In the results section please add in the text the referenced table in parenthesis so readers can check the data in the adequate table. It is impossible now to determine which table are the authors commenting.
Results for Table 2 – The Authors only compared averages without regards for the post-hoc tests. Please take the post-hoc test in consideration and rewrite the results accordingly
Table 3 – This is the TxGxY interaction? In table 4 it is not statistically significant. Also, the results are compared through averages without the regard for post-hoc test. This table should be merged with Table 4 since it is a part of the same ANOVA in the form presented in the significance section of Table 4 (first the main effects, followed by the interactions).
The authors state that there were differences between years in several observed parameters but according to table 4 no significant differences were observed in any of the investigated parameters between the harvest years. How is this possible?
Figure 1. – Is this the same data as in Table 4 but only expressed in percentages?
Chapters 4 and 4.1 should be 3.4 and 3.5
Chapter 5 should be chapter 4
Author Response

(The authors gave the same response as above.)

Round 2
Reviewer 1 Report
Dear Authors:
Thanks for your considerable efforts. All of my comments and notes has been taken into consideration and therefore I find it acceptable for publication after few minor English editing.
Author Response
We thank you for your insightful additional comment, which aimed to further improve the paper. Accordingly, the paper has been carefully revised by an English-speaking colleague. Changes were highlighted in yellow.
Finally we hope the edits made persuade you to accept our manuscript.